# Physiological Features of the Neural Stem Cells Obtained from an Animal Model of Spinal Muscular Atrophy and Their Response to Antioxidant Curcumin

**DOI:** 10.3390/ijms25158364

**Published:** 2024-07-31

**Authors:** Raffaella Adami, Matteo Pezzotta, Francesca Cadile, Beatrice Cuniolo, Gianenrico Rovati, Monica Canepari, Daniele Bottai

**Affiliations:** 1Section of Pharmacology and Biosciences, Department of Pharmaceutical Sciences, University of Milan, Via Balzaretti 9, 20133 Milan, Italy; raffaella.adami@unimi.it (R.A.); pezzottamatteo1999@gmail.com (M.P.); bcuniolo@gmail.com (B.C.); genrico.rovati@unimi.it (G.R.); 2Human Physiology Unit, Department of Molecular Medicine, University of Pavia, Via Forlanini 6, 27100 Pavia, Italy; francesca.cadile01@universitadipavia.it (F.C.); monica.canepari@unipv.it (M.C.)

**Keywords:** spinal muscular atrophy, neural stem cells, neurological disease, curcumin, oxidative stress, NRF2

## Abstract

The most prevalent rare genetic disease affecting young individuals is spinal muscular atrophy (SMA), which is caused by a loss-of-function mutation in the telomeric gene survival motor neuron (*SMN*) *1*. The high heterogeneity of the SMA pathophysiology is determined by the number of copies of *SMN2*, a separate centromeric gene that can transcribe for the same protein, although it is expressed at a slower rate. SMA affects motor neurons. However, a variety of different tissues and organs may also be affected depending on the severity of the condition. Novel pharmacological treatments, such as Spinraza, Onasemnogene abeparvovec-xioi, and Evrysdi, are considered to be disease modifiers because their use can change the phenotypes of the patients. Since oxidative stress has been reported in SMA-affected cells, we studied the impact of antioxidant therapy on neural stem cells (NSCs) that have the potential to differentiate into motor neurons. Antioxidants can act through various pathways; for example, some of them exert their function through nuclear factor (erythroid-derived 2)-like 2 (NRF2). We found that curcumin is able to induce positive effects in healthy and SMA-affected NSCs by activating the nuclear translocation of NRF2, which may use a different mechanism than canonical redox regulation through the antioxidant-response elements and the production of antioxidant molecules.

## 1. Introduction

Young patients suffering from neurological illnesses represent significant financial and emotional costs to our society. These patients require extensive care, and the responsibility typically falls on parents and other family members. This is especially true for those with severe presentations of diseases. Spinal muscular atrophy (SMA) is a dangerous disorder characterized by the death of lower motor neurons. In extreme situations, patients can experience early-life respiratory failure. SMA mainly affects infants, children, and young adults.

The first accounts of SMA date back to the late 1800s and were written by Johann Hoffman and Guido Werdnig [1]. Today, 1 in 6000–11,000 people have SMA [2,3,4,5,6]. According to Sugarman and collaborators [7], the carrier frequency for SMA is 1 out of 47 in Caucasians and 1 out of 72 in African Americans, making it the second most common disease affecting young people after cystic fibrosis.

SMA has a wide range of clinical manifestations, from adult onset to embryonic lethality. The condition can be categorized into eight unique phenotypes [8] depending on the severity and age at onset. Type 0, also known as 1A, is diagnosed during pregnancy when the fetus lessens its movements. After delivery, the patients require urgent breathing assistance and typically die within the first month. Werdnig–Hoffman disease is another severe form of SMA that is sometimes referred to as types 1b and 1c (with 1c slightly less severe than 1b). The disorder first appears before the patient is six months old and is characterized by an inability to sit, which forces the patient to remain prone and makes breathing without assistance difficult. Following therapy, those patients on respiratory support survive for an average of two years. The onset of type 2 a and b SMA is detected at 7–18 months; the patients are able to sit but not walk. This is considered to be an intermediate phenotype, and the patients have an average lifespan of over 40 years.

Those patients with type 3 SMA have a life expectancy similar to the general population. Type 3 is diagnosed by 18 months of age and can be further divided into two categories: 3a, which is characterized by a loss of the existing walking ability, and 3b, characterized by the retention of the walking ability for a longer time than 3a. The patients with type 4 begin to exhibit symptoms in their second and third decades of life, and their prognosis and gait are similar to those of the general population. Type 4 SMA often goes undiagnosed due to its modest traits [8,9]. The most commonly used classification system for SMA is Dubowitz’s decimal subclassification, which enables the categorization of individuals within each type by varying degrees of severity (1.1–1.9 for type 1, 2.1–2.9 for type 2, and 3.1–3.9 for type 3) [10].

SMA is caused, in molecular terms, by the inactivation of the survival motor neuron 1 (*SMN1*) gene, which is located in the unstable telomeric region of chromosome 5. This inactivation can be the result of a single mutation or a substantial deletion [11]. The compensatory activity of another centromeric gene derived from the duplication of the telomeric *SMN1*, survival motor neuron 2 (*SMN2*), is responsible for the high diversity of this disease. *SMN2* can transcribe for the same protein as *SMN1*, albeit with a lower rate of expression due to the non-inclusion of exon 7, which produces a truncated mRNA that is mostly translated into an inactive protein. In *SMN2*, about 10% of the total transcript includes exon 7 and codifies for the full-length protein. In patients who have three or more copies of *SMN2* [11,12], a sufficient amount of full-length protein is produced, and the condition is less severe.

The degree of the reduction in the SMN protein is the main determinant of the disease, although elevated oxidative stress (OS) has also been linked to SMA. Reactive oxygen species (ROS) are important intermediates in cell signaling pathways that are removed by the antioxidant defense system, which includes, among other things, non-enzymatic antioxidants, such as glutathione, uric acid, and coenzyme Q10, and enzymatic antioxidants, such as catalase, superoxide dismutase, and glutathione peroxidase [13]. Tissues with a high oxygen demand are particularly sensitive to OS; the nervous system is the most vulnerable. Therefore, it is possible that high levels of OS could make certain neurological illnesses worse, particularly those that cause deficits in the antioxidant defense system. 

OS can affect cell physiology in a variety of aspects. Singh and collaborators [14] demonstrated that OS can change the splicing in the *SMN1* and *SMN2* genes. The data from a multi-exon skipping detection assay demonstrated a promoter sequence role in the regulation of *SMN* exon 7 splicing under OS conditions; however, the cis elements, within *SMN* exon 7 and/or within the flanking intronic sequences, also contribute towards OS-induced exon 7 *SMN2* splicing. *SMN2* may be able to function as a kind of sensor to identify OS [15]. In addition, mechanistic studies on a cellular model of cancer found that nuclear factor (erythroid-derived 2)-like 2 (NRF2) is able to bind two antioxidant-response sites in the *SMN1* promoter and that NRF2 transcriptionally controls the production of *SMN* mRNA. Physically, NRF2 was linked to the SMN protein post-transcriptionally. Their connection requires the YG box, the area encoded by exon 7 of *SMN*, and the Neh2 domain of NRF2 [16]. Thus, antioxidants may have a regulatory role in SMA.

A potent antioxidant and the major polyphenolic compound from turmeric is curcumin (Cur), a phyto-polyphenol pigment ((1E,6E)-1,7-bis(4-hydroxy-3-methoxyphenyl)hepta-1,6-diene-3,5-dione) (Figure 1).

Its effects have been documented by Eastern nations, such as China and India, for over 4000 years. A diet high in Cur helps to lower inflammation and fight a variety of human diseases, such as diabetes and cancer [17]. Due to its significant pre-systemic metabolism in the liver and gut and its limited water solubility (11 ng/mL at physiological pH), the systemic bioavailability of Cur is less than 1% after an oral dosage in humans. Cur’s substantial metabolism via the first-pass metabolism within the gut microbiota is the primary metabolic route. Studies on the pharmacokinetics of Cur have found several methods for boosting bioavailability, such as the co-administration of the glucuronidation inhibitor piperine, bypassing the first-pass metabolism, non-oral delivery methods, and delivery methods based on nanotechnology [17,18]. Pilot clinical trials (ID numbers NCT00164749 and CT01716637) suggest that Cur is safe and effective in Alzheimer’s disease patients and the general population. The participants in this clinical experiment were administered 1 or 4 g of Cur per day, either as a powder to add to meals or in capsules. Cur was shown to reach concentrations that are likely sufficient for therapeutical purposes in the brains (0.41 μg/g) and plasma (0.60 μg/g) of mice [18]. 

In this study, we describe the stemness properties, such as proliferation, self-maintenance, and metabolic activity, of the neural stem cells (NSCs) derived from the sub-ventricular zone (SVZ) of *SMN*-Δ7 mice in order to describe their physiological features. We also assess the impact of Cur supplementation on the physiological features of NSCs and SMN and NRF2 levels. 

## 2. Results

### 2.1. Proliferation Capability

The NSCs from eleven-day-old *SMN* mice demonstrated a reduced proliferation capability compared to matched-age NSCs from non-congenic (wild-type [WT]) mice. We compared the proliferation of the NSCs by constructing growth curves to analyze the cell growth during the exponential growth phase, then compared the overall slopes of the growth curves (Figure 2A; Appendix A). 

The average slope and the standard error of the mean (SEM) were, respectively, 0.1213 ± 0.01212 (N = 5) for the WT-NSCs (WT-ctr) and 0.0818 ± 0.00853 (N = 5) for the SMA-NSCs (SMA-ctr). The values were significantly different (*p* = 0.0285), and we found that the NSCs from the SMA mice have growth curve slopes that are 33% lower than WT mice, indicating a lower proliferation rate. There was also a significant difference in the doubling time, which is 151% higher in SMA-ctr (Figure 2B and Appendix A). The WT-ctr population doubled in 2.4476 ± 0.2012 days (N = 5) vs. 3.698 ± 0.5424 days (N = 5), *p* = 0.0313, for SMA-ctr. These statistically significant differences indicated some relevant impairments in the important physiological features of the NSCs. 

### 2.2. Clonogenic Capability 

A peculiar feature of NSCs is their ability to self-maintain; this can be evaluated by measuring the clonogenic capacity of the cells. In order to assess the clonogenic capacity of the cells, we took measures to reduce the possibility that a single clone (the neurosphere) could be produced by more than one NSC. The NSCs were plated by serial dilution, with one cell per well in a 96-well multi-well plate. Then, 3–5 days later, we counted the neurospheres that formed. The ratio between the number of neurospheres formed and the cells plated provides us the clonogenic capability of the NSCs in terms of the percentage of the total cells plated. The NSCs from the SMA mice demonstrated a 45% lower capability to produce clones than the WT-NSCs (Figure 2C), 10.33 ± 1.022 (N = 6) for WT-ctr and 5.667 ± 1.022 (N = 6) for SMA-ctr, *p* = 0.0090 (Figure 2C and Appendix A). Clonogenicity is particularly important in the early phases of development, and these statistically significant variations suggest that the NSCs derived from the SMA mice are impaired in terms of self-maintaining.

These experiments were performed using the cells obtained from the growth curve analysis; they were collected and plated at the start of the growth curve and again at the end of the growth curve (about 50 days later). The results at these two time points were comparable (Appendix A).

### 2.3. Expression of Stemness Neural Markers

The NSCs obtained from the SMA and WT animals expressed typical markers such as Nestin and SOX2 (Appendix A). No significant differences in the expression of the stemness neural markers were found between the SMA- and WT-NSCs (Appendix A). This study examined a few stemness genes for NSCs; the results showed that neither SOX2 nor Nestin differed between the NSCs obtained from SMA and those derived from WT mice, suggesting that the primary stemness expression characteristics were retained.

### 2.4. Metabolic Activity

The 3-(4,5-Dimethylthiazol-2-yl)-2,5-Diphenyltetrazolium Bromide (MTT) assay did not find significant differences in the mitochondrial metabolic activity between the WT- and SMA-NSCs (Appendix A).

### 2.5. SMN Protein Levels

In order to confirm that SMA-ctr expresses a low amount of the SMN protein, as expected from the previous studies [19], we prepared NSC proteins by harvesting them as neurospheres. The proteins were separated by electrophoresis (based on their molecular weight) and transferred to the polyvinylidene difluoride (PVDF) membrane in order to perform an immunoblot. We evaluated the total proteins using SYPRO^®^ Ruby Protein Blot Stain because this method is more sensitive than Ponceau Red (or other staining techniques), and it is compatible with immune staining. The proteins were quantified to normalize the immunoblot results by the total proteins present in the membrane [20], and the numbers reported below are expressed in arbitrary units. 

SMA-ctr has a significantly lower expression of the SMN protein (11.4%) compared to WT-ctr (Figure 3 and Appendix A). Indeed, the results are 0.002869 ± 0.0002887 (N = 3) for WT-ctr and 0.000328 ± 0.0001083 (N = 4) *p* < 0.0001 for SMA-ctr. As expected from the animal model used, the production of the SMN protein is drastically lower in the NSCs obtained from the SMA animals in comparison to the NSCs from the WT mice. Indeed, in this mouse model, the *Smn* gene was eliminated and substituted with the human *SMN2* gene, so we expected to obtain a very reduced production of the SNM protein (Shafey et al., 2008 [21]).

### 2.6. NRF2 Analysis

No significant differences were found in the SMA-ctr and WT-ctr expression of NRF2. Therefore, the NRF2 nuclear levels were evaluated. Four cultures for each group were studied, and more than 100 cells per culture were analyzed. We found a significant reduction in the ratio of nuclear NRF2 to cytoplasmic NRF2. The ratio of nuclear/cytoplasmic NRF2 in SMA-ctr is 34% lower than the ratio in WT-ctr (Figure 4 and Appendix A), 2.306 ± 0.040 (N = 4) for WT-ctr and 1.522 ± 0.023 (N = 4) *p* < 0.0001. 

### 2.7. Curcumin Treatment of NSCs

As previously mentioned, we tried to counteract the oxidative state present in SMA cells to assess whether a reduction in the OS can reverse some of the detrimental aspects of SMA.

We evaluated the concentration of Cur that exerts a significant increase in the proliferation rate of the SMA-NSCs (SMA-Cur). Since the proliferative and self-maintenance capabilities of the NSCs were affected in SMA-Cur, we evaluated different dosages of Cur ranging from 0.1 μM to 50 μM on NSCs, and we found that the treatment induced a significant increase in the proliferation at a concentration of 0.5 μM and a toxic effect at concentrations over 10 μM (Appendix A). 

### 2.8. Curcumin Increases Proliferation Capability, Doubling Time, and Self-Maintenance

On the basis of the dose–response experiments, we decided to treat different NSC cultures with 0.5 μM of Cur. We found that the vehicle, dimethyl sulfoxide (DMSO) (1:1000), did not affect the NSC proliferation (Appendix A). Five vehicle-treated SMA-ctr cultures (the control) were compared to the same five cultures treated with Cur. The Cur treatment increased the proliferation rate (slope of the growth curve) in SMA-Cur by 16% compared to the vehicle-treated cells from SMA-ctr (Figure 5A; Appendix A), 0.0818 ± 0.008533 (N = 5) for SMA-ctr-NSCs and 0.09486 ± 0.008893 (N = 5) *p* = 0.0043 for SMA-Cur-NSCs. 

The treatment with Cur also significantly reduced the doubling time (Figure 5B and Appendix A) in the SMA-NSCs by about 20% more than the vehicle-treated cells. The doubling time of the SMA-ctr-NSCs was 3.698 ± 0.5424 days (N = 5) and 2.067 ± 0.3966 days (N = 5) *p* = 0.0231 for SMA-Cur (Appendix A). 

Moreover, 0.5 μM of Cur increased the clonogenic capability of SMA-Cur by 50% compared to the vehicle-treated SMA-NSCs, 5.667 ± 1.022 (N = 6) for SMA-ctr and 8.333 ± 1.202 (N = 6) in SMA-Cur, *p* = 0.0005 (Figure 5C and Appendix A). The Cur treatment was able to significantly enhance the proliferation, doubling time, and self-renewal of the NSCs obtained from the SMA mice in comparison to the untreated SMA-NSCs. This result is quite remarkable, as shown in Figure 5. These levels of proliferation, doubling time, and self-renewal are quite similar to those of the WT-untreated-NSCs.

### 2.9. Curcumin Does Not Alter the Expression of Stemness Markers

No significant differences regarding the Nestin or SOX2 expression were found after the SMA- and WT-NSCs were treated with Cur (Appendix A). After receiving the Cur therapy, the expression of two crucial NSC stemness markers, Nestin and SOX2, remained unchanged. It is quite likely that neither the illness nor a nutritional treatment will be able to change the expression of these two proteins (see Section 2.3).

### 2.10. Curcumin Significantly Increases Metabolic Activity 

The Cur treatment significantly increases the mitochondrial metabolic activity in SMA-Cur by 16% more than SMA-ctr (Appendix A). The metabolic activity was 0.09227 ± 0.01290 (arbitrary units) for SMA-ctr and 0.1125 ± 0.01426 *p* = 0.0272 for SMA-Cur. As expected, Cur significantly changes the metabolism of the NSCs derived from SMA mice, as measured by the MTT assay. No effects of Cur were detected for the WT-NSCs.

### 2.11. Curcumin Significantly Increases the Expression Levels of SMN

After normalizing our data for the total protein present in the cells, we found that the expression of SMN in the SMA-Cur-NSCs significantly increased by about 120% more than the vehicle-treated SMA-NSCs (Figure 6 and Appendix A), 0.000328 ± 0.0001083 (N = 6) for SMA-ctr and 0.0007265 ± 0.0002371 (N = 6) for SMA-Cur, *p* = 0.0475. The SMN protein expression level was significantly increased by the Cur treatment in the SMA-NSCs. This effect, however, was not sufficient to reach the expression level of the WT-NSCs.

### 2.12. Curcumin Significantly Increases the Nuclear/Cytoplasmic NRF2 Ratio

We found a significant reduction in the ratio of the nuclear Nrf2 to cytoplasmic NRF2. SMA-Cur has a significantly higher nuclear/cytoplasmic NRF2 ratio of 5% compared to SMA-ctr (Figure 7 and Appendix A), 1.598 ± 0.021 (N = 4) for SMA-Cur and 1.522 ± 0.023 (N = 4) *p* = 0.0464 for SMA-ctr. These findings showed that, as compared to untreated SMA-NSCs, the Cur therapy is able to greatly increase the nuclear translocation of NRF2 in the NSCs derived from SMA mice.

## 3. Discussion

A reduced level of SMN protein induces the onset of SMA. This pathology mostly affects nervous tissue, although its more aggressive forms can also damage the peripheral tissues and systems [22,23]. The therapeutic approaches, mainly developed in the last decade, which include Spinraza (Nursinersen), Onasemnogene abeparvovec-xioi (Zolgensma), and Evrysdi (Risdiplam), have contributed to a completely new version of the disease, with a change in the phenotype of the patients after their treatments [24]. These approaches, however, leave many patients untreated or with poor outcomes if they have already lost too many motor neurons, so other interventions are desirable. In fact, most of the treatments for SMA are less effective if administered outside the ideal time window. Furthermore, since a large portion of the growth of motor neurons and systemic pathomechanistic alterations take place in the womb, postnatal therapy may not be able to reverse the irreparable damage.

Given the growing body of research [25] demonstrating mitochondrial dysfunction and oxidative stress in the pathophysiology of SMA, an antioxidant therapy may improve the disease’s overall prognosis. Moreover, 11-day-old *SMN*-Δ7 mice have evident pathological signs at the muscular level, including reduced diaphragmatic functionality and muscular mitochondrial and redox impairment [26]. Other neurological illnesses have benefited from strategies that increase mitochondrial biogenesis [27,28,29]. For instance, there is evidence that antioxidants such as ergothioneine are beneficial for some neurological illnesses and healthy aging [30].

In this work, we demonstrated that Cur has a beneficial effect on different features of NSC physiology. The treatment of the NSCs isolated from the SVZ of SMA (*SMN*-Δ7) and healthy mice with Cur improved the proliferation, clonogenicity, and SMN expression, and partially recovered the differences between the WT-NSCs and SMA-NSCs.

Our first goal was to evaluate the physiological characteristics of the NSCs from SMA mice. We examined the proliferation capability of the NSCs and found a significantly lower proliferation rate, doubling time, and clonogenic capability compared to the WT mice. In an outstanding paper by the Kothary group [21], it was demonstrated that the NSCs from *SMN*^−/−^ and *SMN2* mice (obtained from striata of E14.5 embryos), which represent a model of very severe SMA, did not show remarkable differences in proliferation from the WT animals. In another paper by Luchetti and collaborators [31], *SMN*-Δ7 stem cells were derived from E13.5 spinal cords. These cells showed an increased metabolism that the authors described as proliferation in the first few days after plating [31]. In our long-term proliferation and growth curve experiments, we found a significant reduction in the proliferation capability of the SMA-NSCs (Figure 2A; Appendix A). We can hypothesize that these differences are due to the differing mouse models, cell origins, and embryonic striata [21]. We used adult SVZ in addition to embryonic spinal cords. Moreover, the method of analysis of proliferation was different. Luchetti and collaborators [31] analyzed the viability by conducting an MTT assay within three days; in contrast, we carried out a dynamic study of the proliferation for more than 50 days. In our analysis, we also found that the viability was not significantly different between the SMA-NSCs and WT-NSCs (Appendix A). 

To test the efficacy of the Cur treatment, we first analyzed the effect of this nutraceutical compound on the proliferation capability of the NSCs; as a test, we used WT-NSCs, and we found that there was a significantly increased proliferation of the cells at a concentration of 0.5 μM. However, concentrations of 10 μM and higher induced a toxic effect (Appendix A). These results are in concordance with many previous works [32,33,34,35,36].

Using 0.5 μM of Cur dissolved in DMSO as a vehicle (with a final concentration of 1:1000), we performed all the further experiments after ensuring that a 1:1000 concentration of DMSO did not change the features of the NSCs (Appendix A).

We performed a growth curve assay by counting the number of cells during the exponential growth passages to determine the effect of Cur. This process demonstrated that the Cur treatment significantly increased the proliferation rate (Figure 5A; Appendix A); the proliferation capability of SMA-ctr was significantly lower than the proliferation capability of WT-ctr (Figure 2; Appendix A). The treatment with Cur increased the differences in the proliferation rates to non-significant levels, which indicates the great potential of this nutraceutical compound to stimulate recovery in SMA-NSCs. This recovery is also demonstrated in other physiological characteristics of NSCs, such as doubling time and clonogenic capacity (Appendix A). The WT-Cur-NSCs showed a significantly higher proliferation rate than SMA-Cur (Appendix A), indicating that Cur is able to increase the proliferation rate in healthy cells as well, as reported in other studies [32,33,34,35,36]. All these data indicate that Cur is able to restore the proliferation, doubling time, and self-maintenance almost at the same levels as the WT-ctr untreated NSCs (Appendix A). Moreover, we noticed that the Cur treatment was also able to significantly modify the behavior of the WT cells (Appendix A), increasing the proliferation rate, reducing the doubling time, and increasing the self-maintenance.

Additionally, we found that NSCs express stem cell markers such as Nestin and SOX2. We did not find significant differences between the WT- and SMA-NSCs. The treatment with Cur did not have any effect on the expression of these markers (Appendix A). 

The finding that the SMN protein expression is significantly lower in SMA-NSCs with respect to WT-NSCs was expected. The untreated SMA-NSCs expressed only 11.4% of the SMN proteins produced by the WT-NSCs (Figure 2B of Le paper) [19]. The SMA-NSCs increased the level of the SMN proteins by 221% (Figure 6 and Appendix A). Although this was a high increase, it was not enough to restore normal levels of the expression of SMN proteins; it represents about 25% of the protein levels present in WT-ctr. 

In order to study the OS levels of these cells, we analyzed the NRF2 expression using an immunocytochemistry approach. While no differences were detected in the whole-cell NRF2 of the SMA-NSCs compared to the WT-NSCs, a significantly lower expression was observed in the ratio of the nuclear and cytoplasmic NRF2 in the SMA-NSCs. Increased ROS levels were found in several studies using cellular models and post-mortem tissues from SMA patients, both before and after the beginning of SMA symptoms [37,38,39], and NRF2 plays a pivotal role in ROS regulation.

We attempted to balance the oxidative state found in the SMA cells to determine whether lowering the OS may reverse the effects of SMA. The Cur treatment induced a significant although slight recruitment of NRF2 in the nucleus; however, this result clearly indicated that the effect of Cur is not sufficient to restore a low level of OS (Appendix A).

The result that Cur induced a significant increase in the SMN protein expression and, although low but significant, an increase in the NRF2 expression in the SMA-NSCs prompted us to study the downstream mechanisms of the NRF2 pathway. As the next step, we investigated the downstream genes in the NRF2 pathway, namely NQO1, by the immunocytometry approach. However, we did not find any significant differences before and after the Cur treatment. We were surprised by this result, yet it is known that NRF2 can act through different pathways in addition to the one involving NQO1. 

In a recent paper, Cui and collaborators [16] demonstrated that NRF2 was able to positively modulate the SMN expression in non-small-cell lung cancer. The study also identified two potential NRF2 binding sites (ARE, antioxidant-responsive elements) within 1 kb of the *SMN1* promoter [16]. Our mouse model [19] contains all the SMN2 human gene, and the SMA cDNA lacks exon 7, which is driven by a 3.4 kb SMN human promoter fragment. Small differences were detected between the telomeric (*SMN1*) or centromeric (*SMN2*) promoters [40,41], so both contain the two potential AREs, but the *SMN2* promoter has almost half the transcriptional activity of the telomeric promoter.

We can hypothesize, with a mechanistic model, that Cur could increase the levels of the SMN protein, as described in Figure 8. 

Cur can modify the Keap1 Cys151 residue and inhibit the ability of the Cullin3–Rbx1 E3 ubiquitin ligase complex to target Nrf2 [42]. This will enable an increase in the translocation of NRF2 into the nucleus (Figure 7 and Appendix A) and grow the binding to the ARE sequence that is present in the 3.4 kb SMN human promoter fragment or in the human *SMN* gene inserted in this transgenic animal [16]. Consequently, Cur can induce an increase in the levels of the SMN protein in SMA animals (Figure 6). However, the small increase in SMN due to the curcumin treatment could be attributed to the low efficiency of the *SMN2* human promoter with respect to the *SMN1* promoter [16] that is present in the SMA mouse. 

On the other hand, in WT animals, the NRF2 ratio level is already quite high in comparison to SMA animals, and Cur is not significantly able to modify its translocation into the nucleus. Moreover, it is not yet clear if AREs are present in the mouse *Smn* promoter; the mouse Smn promoter region is quite different from the human one. By using a searching tool for the putative transcription binding site, we found that some putative AREs were present in the *Smn* mouse promoter, although we are not certain that these sites could really bind NRF2 (https://alggen.lsi.upc.es).

In addition, NRF2 directly interacts with the C-terminal SMN and is a part of the Cajal body complex with SMN, Gemini2, and coilin and most likely plays a role in splicing regulation [16].

As already mentioned, Cur enhances NRF2 expression and stability (through the inhibition of KEAP1) [42] and promotes the migration of NRF2 to the nucleus, activating NRF2 downstream targets that are important to prevent oxidative stress-inhibiting inflammatory mediators in many different cellular models, such as macrophage cells [43]. Additionally, the Cur treatment changed the methylation status of the NRF2 gene’s CpG promoter region, causing NRF2 and its target gene, NQO-1, to be expressed again and thus having a chemopreventive impact on prostate cancer [44]. To learn more about the molecular processes behind Cur’s effects on NRF2, in silico investigations like molecular docking and molecular dynamics simulations may be important in the following steps of this research [45,46]. We do not know yet if the NRF2 in our model has a similar function as the one found in non-small-cell lung cancer. However, we think that NRF2 regulation could play a pivotal role in the treatment of SMA, at least in terms of maintaining the residual motor neurons that are still alive in the late phases of the disease. Indeed, many studies report the role of NRF2 in the regulation of the NSC processes, including proliferation, self-renewal, stemness markers, and neuronal differentiation, which are linked to the increase in the SVZ NRF2 levels (which decrease during aging) [47,48]. These effects are critically important during the early phases of development.

## 4. Materials and Methods

Cur phospholipids were obtained and prepared following the protocol outlined by Starvaggi Cucuzza and collaborators [49] with slight modifications. Briefly, Cur was dissolved in DMSO at a concentration of 50 mM and then diluted in the culturing medium at the maximum concentration of DMSO of 1:1,000.

### 4.1. Mouse Models

*SMN*-Δ7 mice were used. *SMN*-Δ7 mice (FVB·Cg-Grm7^Tg(*SMN2*)89Ahmb^ Smn1^tm1Msd^ Tg(*SMN2**delta7)4299Ahmb/J, Strain #: 005025, The Jackson Laboratory) [19] are currently the SMA mouse line that is most frequently utilized in various labs across the globe. Their median survival is roughly 13 days on average. The neuropathology and symptoms of the *SMN*-Δ7 mouse model are comparable to those of type 2 SMA in humans. *SMN*-Δ7 and control littermate (11-day-old mice) were used in our analysis. This age was chosen as a compromise to have a higher number of living SMA animals. In addition to having a longer lifespan, non-congenic mice reflect the diverse genetic backgrounds found in the human population. A neck cut was used to euthanize animals. After removing the brain, the SVZ or other neurogenic areas were cut open for the preparation of NSCs. All procedures were approved by the University of Pavia’s Animal Care and Use Committee in 2021 and still active (protocol reference n° 280/2021-PR) and, in conformity with Italian law, were communicated to the Ministry of Health and local authorities. 

### 4.2. Isolation and Culture of NSCs

The methodology followed the procedures detailed in our group’s earlier publications [50,51]. Briefly, the SVZ-containing tissues were removed along with the brains. A single mouse brain served as the source for each culture. Dissected tissue was stored at 4 °C in a phosphate buffer 0.01 M solution containing 100 U/mL of penicillin, streptomycin (Invitrogen, San Diego, CA, USA), and glucose (0.6%). Because the tissues were soft due to the age of the mice, enzymatic dissociation at 37 °C was not required.

After the tissues were mechanically broken and centrifuged, single cells were extracted [52]. The pellet [53] was re-suspended in proliferation media (PM), and the supernatant was disposed of. Under these circumstances, the tissue’s NSCs produced spheroidal structures (neurospheres) after 5–7 days. These were removed, mechanically separated, and re-plated in PM at a density of 10,000 cells/cm^2^. 

### 4.3. Growth Analysis

Growth curves were obtained from 10 cultures (5 WT and 5 SMA) starting from the third passage. Cells were mechanically dissociated at each passage when the neurospheres reached an appropriate size (around 0.1 mm) and plated in a 6-well plate well at a density of 10,000 cells per cm^2^. Multiplying the proliferation rate (number of viable cells harvested/number of inoculum cells) by the cumulative total number of cells from the preceding passage yielded the cumulative total number of cells for each passage [52,54]. Slopes were calculated as the linear regression of the log(number of cells), implying that the starting point is Log(90,000), where 90,000 is the number of cells plated in this step [52].

The algorithm supplied by http://www.doubling-time.com, was used to calculate the population doubling time [52] using the following formula: DoublingTime = duration ∗ log(2)/(log(FinalConcentration) − log(InitalConcentration)).

### 4.4. Clonal Analyses

The clonal analysis was based on the study of clones derived from single cells [53,55,56], which provides extremely valuable information regarding cell characteristics and enables the creation of NSC lineages. A 96-well multi-well plate containing PM medium was seeded with 50 cells (in the first column, eight wells). Cells were then serially diluted into the following six columns until there was only one cell per well. To verify the number of cells, the cells were recounted so that we could rule out the formation of clones from two or more cells and the effect of cell density. We evaluated the clone ratio at different dilutions. Neurospheres that developed after the cells were left in PM medium for five to seven days were counted. The ratio of the number of neurospheres at the end to the number of cells plated at the beginning was used to determine clonogenicity. The ratio of clonogenicity to the number of cells plated per 100 was then used to compute the percentage of clonogenicity.

### 4.5. MTT Assay

Cells were plated onto Cultrex-coated 96-well plates (TEMA Ricerca, Bologna, Italy) at a concentration of 10,000–15,000 cells/well in 200 μL of the PM at 37 °C and 5% CO_2_. The tetrazolium dye, or MTT (5 mg/mL in phosphate-buffered saline [PBS]; Sigma, Saint Louis, MO, USA), was added to the medium one day after plating and three hours before collection (at a final concentration of 500 μg/mL). After incubating for three hours at 37 °C, the PM was carefully disposed of without dislodging the neurospheres, which were lightly attached to the surface of the well. To lyse the cells, 100 μL of DMSO was added. MTT reduction was evaluated spectrophotometrically using an iMark Microplate Absorbance Reader (Bio-Rad, Hercules, CA, USA) at a wavelength of 550 nm after 15 min at room temperature [52,57]. 

### 4.6. Undifferentiated Cell Immunofluorescence

A total of 20,000 cells were plated into a 48-multi-well plate that held one round glass coverslip (10 mm Cultrex-coated, TEMA Ricerca, Bologna, Italy) in PM medium for 45 min (the minimum duration required for cell attachment) [57]. After that, the medium was removed, and the cells were fixed with 4% paraformaldehyde (PFA) for three minutes at room temperature. PFA was replaced with PBS, and the samples were kept at 4 °C for up to 2 months until immunostaining. The primary antibodies used were mouse anti-Nestin monoclonal antibody (1:300, Immunological Sciences MAB-11004), rabbit anti-NRF2 polyclonal antibody (1:200, Immunological Sciences AB-84017), and rabbit anti-SOX2 polyclonal antibody (1:300, Immunological Sciences MAB-81156). For intracellular epitope immunostaining, the cells were permeabilized with 0.1% triton X100. Secondary antibodies were conjugated with three different fluorophores, Alexa-fluor 555 (goat-anti mouse IgG IS20030; 2 μg/mL), Alexa-fluor 633 (goat anti-mouse IgG IS20121, 2 μg/mL), and Alexa-fluor 488 (goat anti-rabbit IgG IS20012; 2 μg/mL), at a dilution of 1:800. Cell nuclei were stained with 4′,6-diamidino-2-phenylindole (DAPI) at a concentration of 1:1000; subsequently, following PBS washes, the round coverslips were mounted on glass slides using 1,4-diazabicyclo[2.2.2]octane (DABCO) as a mounting medium to prepare for image acquisition. Using a Zeiss LS900 confocal microscope, images of the samples were obtained at 10× and 40× oil objectives, by means of 405, 488, and 630 nm lasers, using Zen Zeiss software (version 3.0.79.0006) Oberkochen, Germany, with acquisition parameters held constant throughout each experimental series.

### 4.7. NRF2 Staining

Images were analyzed with Fiji (version 2.14.0/1.54) [58] and transformed into max projection. Threshold values were derived from primary antibody omission images and applied to obtain a cell mask. The intensity was determined by applying the cell mask to the image stack, reading a multi-measure of all slices, and exporting the negative images to Excel to average and subtract the background read. To calculate relative intensity values, each data point was divided by the mean value of the weighted control used as the reference.

### 4.8. NRF2 Nuclear Staining

Images were analyzed and acquired as described in the previous paragraph [58]. For the nuclear NRF2, 405-nucleus channel max projections were used to obtain nucleus masks. Multi-measure was then applied to the 405 and 488 channels to measure fluorescence. The three most brilliant DAPI slices were used as nuclear marks, and the NRF2 fluorescence was calculated in corresponding 488 slices, averaged, and subtracted from the background read of the negative images.

Total nuclear fluorescence was subtracted from total cell fluorescence in each slice, averaged, and subtracted from the background read of the negative images to calculate Cytoplasmic NRF2. The nuclear-to-cytoplasmic ratio was calculated for each single cell. For relative intensity values, each data point was divided by the mean value of the WT-ctrl-NSCs used as the reference.

### 4.9. Protein Extraction, Preparation, and Quantification and Western Blot Analysis

Cells were centrifuged at 123× *g* for 10 min. The pellet was washed once with PBS, frozen in liquid nitrogen, and conserved at −80 °C until all the cultures/samples were ready. Radio immunoprecipitation assay (RIPA) buffer was used to extract the proteins from five million cells, using 20 μL of RIPA per one million cells. Samples were kept on ice for 10 min to allow the membrane to fully break and were shaken manually three times. The samples were subsequently centrifuged in a cold chamber for 30 min at maximum speed, and then the supernatant containing the proteins was removed and stored at −80 °C. Proteins were quantified using the Bradford method.

Each lane was loaded with 10–15 μg of denatured proteins [59] and analyzed with 10% sodium dodecyl-sulfate polyacrylamide gel electrophoresis (SDS-PAGE). Further, 3 μL of HyperPAGE Prestained Protein Marker (Meridian Bioscience, Cincinnati, OH, USA) were loaded in the first lane. Using a Trans-blot^®^ Turbo^TM^ Transfer System (Bio-Rad, CA, USA), proteins were transferred to nitrocellulose membranes (using a Trans-blot Turbo RTA transfer kit PVDF 1704272 (Bio-Rad, CA, USA) set to 1.3A-25V for 10 min) and blocked with 5% dry milk (Merck, Darmstadt, Germany) in Tween-Tris Buffer Saline Solution (0.05% Tween 20) at room temperature for 3 h. The membranes were then incubated with antibodies against SMN (NB100-1936 (Novus Biologicals, Littleton, CO, USA)) or other epitopes overnight at 4 °C. After that, membranes were incubated for one hour with the appropriate secondary antibodies (Ab goat anti-mouse IGg, horseradish peroxidase (HRP)-linked antibody 7076, cell signaling, 1:3000; Ab goat anti-rabbit IGg, HRP-linked antibody 7074, cell signaling, 1:3000) coupled to HRP after being washed three times for ten minutes each [59]. The reaction was quantified by using the SuperSignal^TM^ West Femto chemiluminescent substrate (Pierce, Rockford, IL, USA). Densitometric analysis was performed with an Odyssey FC Imager (LI-COR, Lincoln, NE, USA). The data were expressed as relative changes in optical density. In order to compare the proteins present in each lane, we loaded the same quantity of sample (10 or 15 μg) in each lane to compare the proteins present, and we used SYPRO^®^ Ruby Protein Blot Stain for nitrocellulose or PVDF membranes (1703127, Bio-Rad, CA, USA) to assess the total amount of proteins in each lane. Particularly in the typical load range for cell lysate, measuring total proteins using Stain-Free technology enables more dependable loading and transfer control than housekeeping proteins and permits more accurate sample normalization [60,61].

### 4.10. Statistical Analysis

The average ± standard error of the mean (SEM) is used to express all the data. Paired or unpaired *t*-tests and one-way ANOVA were used to analyze the data. At *p* < 0.05, results were deemed statistically significant [52].

## 5. Conclusions

To the best of our knowledge, this is the first paper describing the physiology of the NSCs from the adult SVZ of *SMN*-Δ7 mice. Although the antioxidant Olesoxime (a cholesterol derivative that targets mitochondria) was found to have inadequate long-term results in SMA [62], alternative antioxidant approaches could be undertaken. Cur will not be a substitute for the effective new drugs available for SMA; however, this nutraceutical treatment could help to improve the quality of life of SMA patients. Further analysis will be necessary to untangle the molecular mechanisms involved in Cur’s effects on NRF2 in our SMA model. 

## Figures and Tables

**Figure 1 ijms-25-08364-f001:**
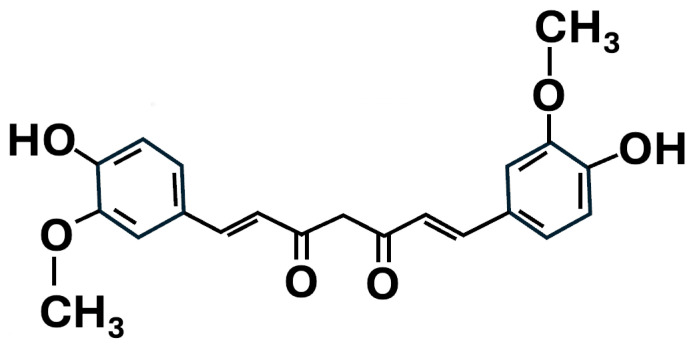
A schematic representation presenting the structural formula of curcumin ((1E,6E)-1,7-bis(4-hydroxy-3-methoxyphenyl)hepta-1,6-diene-3,5-dione).

**Figure 2 ijms-25-08364-f002:**
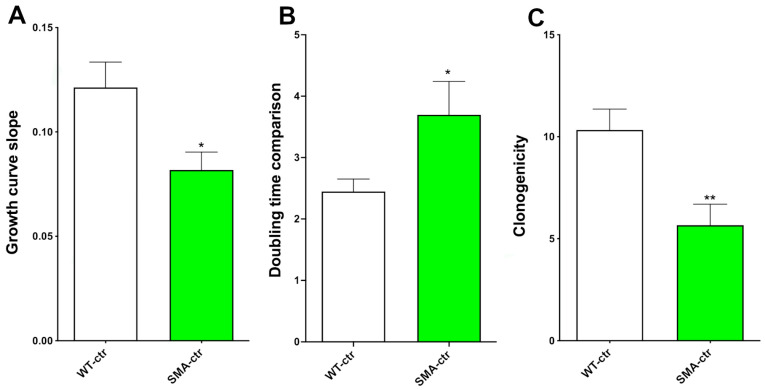
Comparison of growth curve slopes, proliferation doubling times, and clonogenic capability. (**A**) WT-ctr vs. SMA-ctr. The figure represents the comparison between the slopes of the growth curves. White: WT-ctr; green: SMA-ctr; * *p* = 0.0285. Statistical analysis was performed using the two-tailed unpaired *t*-test. (**B**) The average time (in days) required to double the cell population was calculated with non-linear regression analysis. * *p* = 0.0313. Statistical analysis was performed using the one-tailed unpaired *t*-test. (**C**) The clonogenicity of NSCs (their capability to form clones from a single cell) was evaluated through a clonal assay. SMA-ctr showed a reduced clonogenic capability compared to WT-ctr. ** *p* = 0.0090. Statistical analysis was performed using the two-tailed unpaired *t*-test.

**Figure 3 ijms-25-08364-f003:**
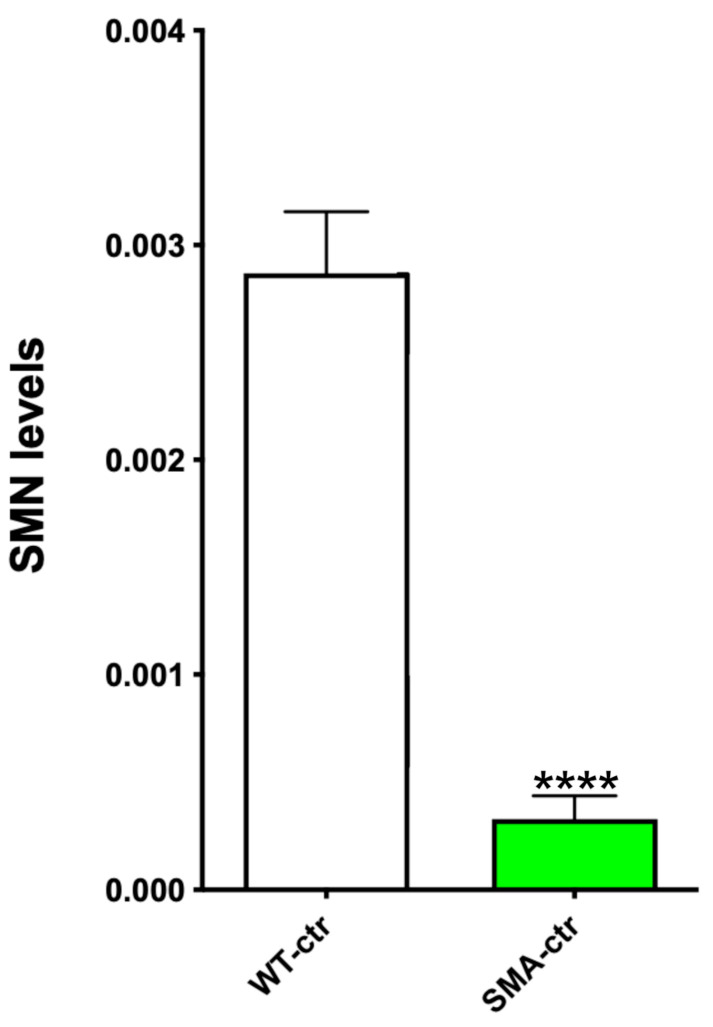
Evaluation of SMA protein levels. Western blot analysis of the SMN protein extracted from WT-ctr and SMA-ctr. Three WT-ctr protein samples and three or four SMA-ctr samples were separated through electrophoresis and transferred to a PVDF membrane. Total protein per lane was measured (as a normalization factor), and SMN protein levels were detected using NB100-1936 Novus Biologicals antibody. White: WT-ctr; green: SMA-ctr. **** *p* < 0.0001. Statistical analysis was performed using the one-tailed unpaired *t*-test.

**Figure 4 ijms-25-08364-f004:**
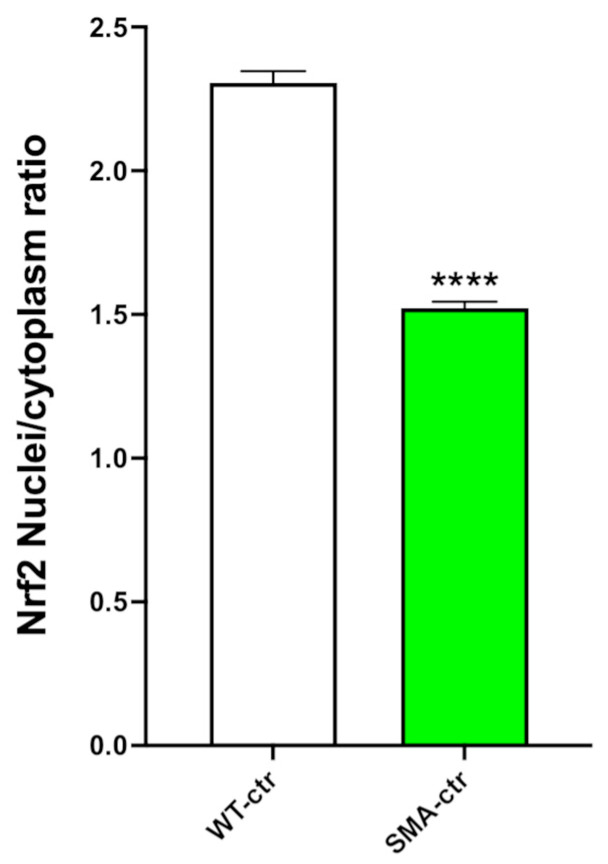
Assessment of NRF2 nuclear/cytoplasmic ratio. We measured the ratio of nuclear and cytoplasmic NRF2 through immunocytochemical analysis. White: WT-ctr; green: SMA-ctr. **** *p* < 0.0001. Statistical analysis was performed using the two-tailed unpaired *t*-test.

**Figure 5 ijms-25-08364-f005:**
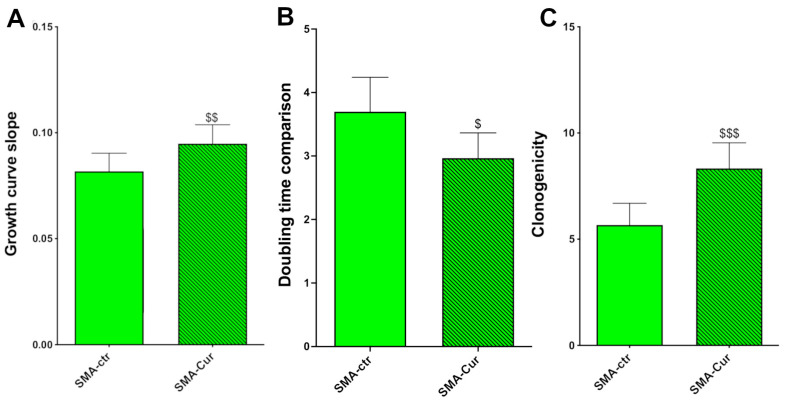
Comparison of growth curve slopes, proliferation doubling times, and clonogenic capability of SMA-NSCs after CUR treatments. (**A**) Comparison of growth curve slopes between SMA-ctr and SMA-Cur. The figure represents the comparison between the slope of the growth curves. ^$$^
*p* = 0.0043. Statistical analysis was performed using the two-tailed paired *t*-test. (**B**) The average time (in days) required to double the cell population was calculated with non-linear regression analysis. ^$^ *p* = 0.0231. Statistical analysis was performed using the one-tailed paired *t*-test. (**C**) Clonogenic capability comparison after Cur treatment. The clonogenicity of NSCs (their capability to form clones from a single cell) was evaluated through a clonal assay. Cur treatments significantly increased the clonal capability compared to SMA-ctr. Green: SMA-ctr; green pattern: SMA-ctr. ^$$$^
*p* = 0.0005. Statistical analysis was performed using the two-tailed paired *t*-test.

**Figure 6 ijms-25-08364-f006:**
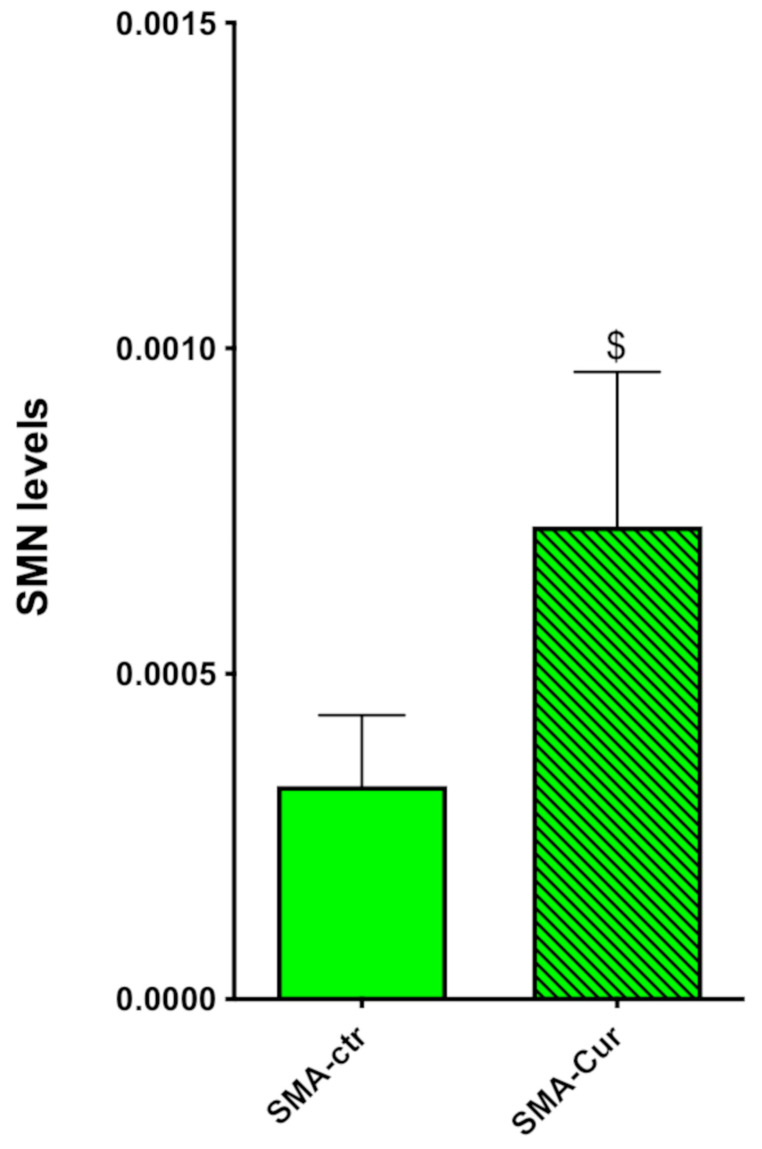
Evaluation of SMA protein levels after Cur treatment. Western blot analysis of the SMN protein extracted from WT-Cur and SMA-Cur. Three WT-Cur protein samples and three or four SMA-Cur samples were separated through electrophoresis and transferred to a PVDF membrane. Total protein per lane was measured (as a normalization factor), and SMN protein levels were detected using NB100-1936 Novus Biologicals antibody. Green: SMA-ctr; green pattern: SMA-Cur. ^$^ *p* = 0.0475. Statistical analysis was performed using the one-tailed paired *t*-test.

**Figure 7 ijms-25-08364-f007:**
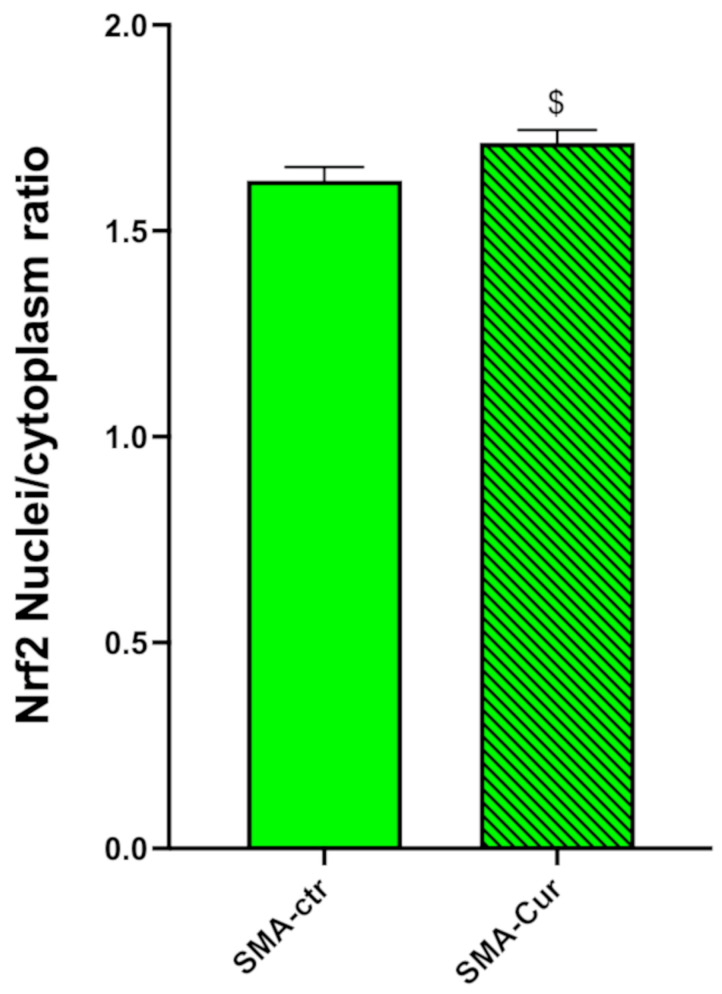
Assessment of the NRF2 nuclear/cytoplasmic ratio. We measured the ratio of nuclear and cytoplasmic NRF2 through immunocytochemical analysis. Green: SMA-ctr; green pattern: SMA-Cur ^$^
*p* = 0.0464. Statistical analysis was performed using the one-tailed paired *t*-test.

**Figure 8 ijms-25-08364-f008:**
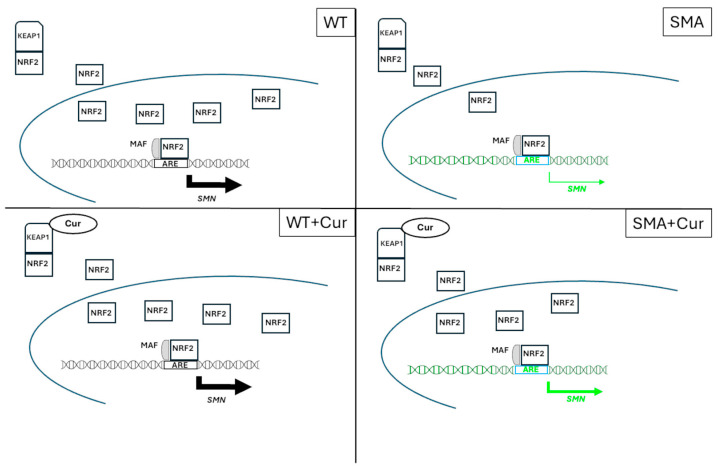
Schematic representation of the proposed mechanism of the Cur effect on SMN expression in WT- and SMA-NSCs.

## Data Availability

Data are available upon request.

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
