# Peer review of "Physiological Features of the Neural Stem Cells Obtained from an Animal Model of Spinal Muscular Atrophy and Their Response to Antioxidant Curcumin"

_ijms, 2024, doi:10.3390/ijms25158364_

Round 1

Reviewer 1 Report

Comments and Suggestions for Authors

In the submitted manuscript Adami et. al. investigated the impact of antioxidant therapy with curcumin on neural stem cells (NSCs) derived from the sub ventricular zone (SVZ) of the survival motor neuron (SMN)-Δ7 mice with the potential to differentiate into motor neurons. The impact of curcumin supplementation on the physiological features of NSCs, SMN, and nuclear factor (erythroid-derived 2)-like 2 (Nrf2) levels was also studied.

The manuscript is relevant to the field. However, there are a few comments that should be addressed before its publication:

Section Introduction

Line 100: Please revise the sentence:« A potent antioxidant extract from turmeric is curcumin (Cur), a phyto-polyphenol pigment ((1E,6E)-1,7-bis(4-hydroxy-3-methoxyphenyl)hepta-1,6-diene-3,5-dione)«. Curcumin is the major polyphenolic compound from turmeric and not an extract.

Line 116: A new figure presenting the structural formula of curcumin would be very beneficial for the readers. 

Section Results

I advise the authors to discuss the obtained results in the corresponding subsections and not in a separate section Discussion. The statistically significant differences in activity after the administration of curcumin should be emphasized and discussed in the main text. All relevant subsections must be numbered.

The statistical tests applied for the analysis of the results should be provided in figure captions and not in the main text.

Figures 1-6: The respective controls should be provided in the figures.

Section Discussion

Line 357: The addition of in silico studies, such as molecular docking and molecular dynamics simulations, would be beneficial to reveal the molecular mechanisms involved in curcumin's inhibitory effects on Nrf2.

References: 

1. Bren et. al. Insight into Inhibitory Mechanism of PDE4D by Dietary Polyphenols Using Molecular Dynamics Simulations and Free Energy Calculations. Biomolecules 2021, 11.3, 479.

2. Tsouri et al. A Monocarbonyl Curcuminoid Derivative Inhibits the Activity of Human Glutathione Transferase A4-4 and Chemosensitizes Glioblastoma Cells to Temozolomide. Pharmaceuticals 2024, 17, 365.

Section Materials and Methods 

The references to the applied protocols should be provided in all subsections.

Comments on the Quality of English Language

The English language is fine, only minor editing is required. 

Author Response

Response to Reviewer 1

Milan 23/07/24

To whom it may concern:

First of all, we would like to express our gratitude for the helpful suggestions the editor and the reviewer gave us. We are convinced that those recommendations will allow us to substantially improve our paper.

Here you can find the changes we made to the manuscript. The changes we made are indicated in blue as insertions and in red as cuts.

We chose to transmit the edits to both reviewers since a significant adjustment was required.

In the submitted manuscript Adami et. al. investigated the impact of antioxidant therapy with curcumin on neural stem cells (NSCs) derived from the sub ventricular zone (SVZ) of the survival motor neuron (SMN)-Δ7 mice with the potential to differentiate into motor neurons. The impact of curcumin supplementation on the physiological features of NSCs, SMN, and nuclear factor (erythroid-derived 2)-like 2 (NRF2) levels was also studied. 

The manuscript is relevant to the field. However, there are a few comments that should be addressed before its publication:

Section Introduction

a)     Line 100: Please revise the sentence:« A potent antioxidant extract from turmeric is curcumin (Cur), a phyto-polyphenol pigment ((1E,6E)-1,7-bis(4-hydroxy-3-methoxyphenyl)hepta-1,6-diene-3,5-dione)«. Curcumin is the major polyphenolic compound from turmeric and not an extract. 

The sentence was changed accordingly:

A potent antioxidant and the major polyphenolic compound extract from turmeric is curcumin (Cur), a phyto-polyphenol pigment ((1E,6E)-1,7-bis(4-hydroxy-3-methoxyphenyl)hepta-1,6-diene-3,5-dione).

b)     Line 116: A new figure presenting the structural formula of curcumin would be very beneficial for the readers. 

A new figure of the structural formula was introduced, the name of the other figures was changed accordingly 

Fig. 1

A schematic representation presenting the structural formula of curcumin ((1E,6E)-1,7-bis(4-hydroxy-3-methoxyphenyl) hepta-1,6-diene-3,5-dione).

c)     Section Results

I advise the authors to discuss the obtained results in the corresponding subsections and not in a separate section Discussion.

We added the discussion in the corresponding subsection

2. 1 Proliferation capability

The average slope and the standard error of the mean (SEM) were respectively 0.1213 ± 0.01212 (N = 5) for WT NSCs (WT-ctr) and 0.0818 ± 0.00853 (N = 5) for SMA NSCs (SMA-ctr). Statistical analysis was performed using the two-tailed unpaired t-test. The values were significantly different (p=0.0285), and we found that NSCs from SMA mice have growth curve slopes that are 33% lower than WT mice, indicating a lower proliferation rate. There is also a significant difference in doubling time, which is 151% higher in SMA-ctr (Fig. 2B and Fig 3 sup). The WT-ctr population doubled in 2.4476 ± 0.2012 days (N = 5) vs. 3.698 ± 0.5424 days (N = 5), p=0.0313 for SMA-ctr. These statistically significant differences indicated some relevant impairments in important physiological features of the NSCs.

2.2 Clonogenic capability

A peculiar feature of NSCs is their ability to self-maintain; this can be evaluated by measuring the clonogenic capacity of the cells. In order to assess the clonogenic capacity of cells, we took measures to reduce the possibility that a single clone (the neurosphere) is produced by more than one NSC. The NSCs were plated by serial dilution, with one cell per well in a 96-well multi-well plate. Then, 3–5 days later, we counted the neurospheres that formed. The ratio between the number of neurospheres formed and the cells plated gives us the clonogenic capability of the NSCs in terms of percentage of the total cells plated. NSCs from SMA mice demonstrated a 45% lower capability to produce clones than WT NSCs (Fig. 2C), 10.33 ± 1.022 (N = 6) for WT-ctr, and 5.667 ± 1.022 (N = 6) for SMA-ctr, p=0.0090 (Fig. 2C and Fig. 4 sup). Clonogenicity is particularly important in the early phases of development, and these statistically significant variations suggest that the NSCs derived from the SMA mice are impaired in self-maintaining.

2.3 Expression of stemness neural markers

NSCs obtained from SMA and WT animals expressed typical markers such as Nestin and SOX2 (Fig. 5 sup). No significant differences in the expression of stemness neural markers were found between SMA and WT NSCs (Fig. 5 sup). This study examined a few stemness genes for NSCs; the results showed that neither SOX2 nor Nestin differed between NSCs obtained from SMA and those derived from WT mice, suggesting that the primary stemness expression characteristics were retained.

2.5 SMN protein levels

In order to confirm that SMA-ctr expresses a low amount of SMN protein, as expected from previous studies (Le et al., 2005), we prepared NSC proteins by harvesting them as neurospheres. Proteins were separated by electrophoresis (based on their molecular weight) and transferred to the polyvinylidene difluoride (PVDF) membrane in order to perform an immunoblot. We evaluated the total proteins using SYPRO® Ruby Protein Blot Stain because this method is more sensitive than Ponceau Red (or other staining techniques) and it is compatible with immune staining. Proteins were quantified to normalize immunoblot results by the total protein present in the membrane (Fosang and Colbran, 2015) and the number reported below are expressed in arbitrary units.

SMA-ctr has a significantly lower expression of SMN protein (11.4%) compared to WT-ctr (Fig. 3 and Fig. 7 sup). 0.002869 ± 0.0002887 (N = 3) for WT-ctr, and 0.000328 ± 0.0001083 (N = 4) p<0.0001 for SMA-ctr. As expected from the animal model used, the production of SMN protein is drastically lower in the NSCs obtained from SMA animals in comparison to NSCs from WT mice. Indeed, in this mouse model, the Smn gene was eliminated and substituted with the human SMN2 gene, so we expect to have a very reduced production of SNM protein (Shafey et al., 2008).

2.8 Curcumin increases proliferation capability, doubling time, and self-manteinance

0.5 μM of Cur increased the clonogenic capability of SMA-Cur by 50% compared to vehicle-treated SMA NSCs, 5.667 ± 1.022 (N = 6) for SMA-ctr, and 8.333 ± 1.202 (N = 6) in SMA-Cur, p=0.0005 (Fig. 5 C and Fig. 4 sup). Cur treatment was able to significantly enhance the proliferation, doubling time, and self-renal of NSCs obtained from SMA mice in comparison to untreated SMA NSCs. These results are quite amazing, as shown in Fig. 4. These levels of proliferation, doubling time, and self-renewal are statistically similar to those of WT-untreated NSCs.

2.9 Curcumin does not alter the expression of stemness markers

No significant differences of Nestin or SOX2 expression were found after SMA and WT NSCs were treated with Cur (Fig. 5 sup). After receiving Cur therapy, the expression of two crucial NSC stemness markers, Nestin and SOX2, remains unchanged. It's quite likely that neither the illness nor nutritional tretment will be able to change the expression of these two proteins (see section 2.3).

2.10 Curcumin significantly increases metabolic activity

Cur treatment significantly increases the mitochondrial metabolic activity in SMA-Cur by 16% more than SMA-ctr (Fig. 6 sup). The metabolic activity was 0.09227 ± 0.01290 (arbitrary units) for SMA-ctr, and 0.1125 ± 0.01426 p= 0.0272 for SMA-Cur.  As expected, Cur significantly changes the metabolism of the NSCs derived from SMA mice. As expected, Cur significantly changes the metabolism of the NSCs derived from SMA mice, as measured by the MTT assay. No effects of Cur were detected for WT NSCs.

2.11 Curcumin significantly increases the expression levels of SMN

After normalizing our data for the total protein present in the cells, we found that the expression of SMN in SMA-Cur NSCs significantly increased by about 120% more than vehicle-treated SMA NSCs (Fig. 6 and Fig 7 sup), 0.000328 ± 0.0001083 (N = 6) for SMA-ctr, and 0.0007265 ± 0.0002371 (N = 6) for SMA-Cur, p=0.0475. The SMN protein expression level was significantly increased by Cur treatment in SMA NSCs. This effect, however, was not sufficient to reach the expression level of the WT NSCs.

2.12 Curcumin significantly increases the nuclear/cytoplasmic NRF2 ratio

We found a significant reduction in the ratio of nuclear NRF2 to cytoplasmic NRF2. SMA-Cur has a significantly higher nuclear/cytoplasmic NRF2 ratio of 5% compared to SMA-ctr (Fig. 7 and 8 sup), 1.598 ± 0.021 (N = 4) for SMA-Cur, and 1.522 ± 0.023 (N = 4) p=0.0464 for SMA-ctr. These findings showed that, as compared to untreated SMA NSCs, Cur therapy is able to greatly increase the nuclear translocation of NRF2 in NSCs derived from SMA mice.

d)     The statistically significant differences in activity after the administration of curcumin should be emphasized and discussed in the main text.

The statistic significalce was introduced in the discussion.

We attempted to balance the oxidative state found in SMA cells to determine whether lowering OS may reverse the effects of SMA. Cur treatment induced a significant although slight recruitment of NRF2 in the nucleus, however, this result clearly indicated that the effect of Cur is not sufficient to restore a low level of OS (Fig. 8A supp).

This recovery is also demonstrated in other physiological characteristics of NSCs such as doubling time and clonogenic capacity (Fig. 3 sup and 4 sup). WT-Cur NSCs showed a significantly higher proliferation rate than SMA-Cur (Fig. 2 sup), indicating that Cur is able to increase the proliferation rate in healthy cells as well, as reported in other studies (Kim et al., 2008; Son et al., 2014; Attari et al., 2015; Ma et al., 2018; Calabrese et al., 2019). All these data indicated that Cur is able to restore proliferation, doubling time, and self-maintenance almost at the same levels as the WT-ctr untreated NSCs (Figs. 2 supp, 3, supp, and 4 supp). Moreover, we noticed that the Cur treatment was also able to significantly modify the behavior of the WT cells (Figs. 2 supp, 3, supp, and 4 supp) increasing the proliferation rate, reducing the doubling time. and increasing the self-manteinance.

We attempted to balance the oxidative state found in SMA cells to determine whether lowering OS may reverse the effects of SMA. Cur treatment induced a significant although slight recruitment of NRF2 in the nucleus, however, this result clearly indicated that the effect of Cur is not sufficient to restore a low level of OS (Fig. 8A supp).

The result that Cur induces a significant increase in SMN protein expression and, although low but significant, an increase in NRF2 expression in SMA-NSCs prompted us to study downstream mechanisms of the NRF2 pathway. As the next step, we investigated downstream genes in the NRF2 pathway, namely NQO1 by immunocytometry approach. However, we did not find any significant differences before and after Cur treatment. We were surprised by this result; yet, it is known that NRF2 can act through different pathways in addition to the one involving NQO1.

e)     All relevant subsections must be numbered.

We inserted the subsection numbers

f)      The statistical tests applied for the analysis of the results should be provided in figure captions and not in the main text. 

The statistical tests were moved in the figure caption.

g)     Figures 1-6: The respective controls should be provided in the figures.

All figures include the appropriate controls WT-ctr in paragraphs: 2.1, 2.2, 2.5, 2.6 and in figures 2-4; and SMA-ctr in paragraphs 2.8, 2.11, 2.12 and in in figures 5-7. Supplemental figures include overall graphs to summarize all the findings andx comparison.

h)     Section Discussion

Line 357: The addition of in silico studies, such as molecular docking and molecular dynamics simulations, would be beneficial to reveal the molecular mechanisms involved in curcumin's inhibitory effects on NRF2.

A new sentence was introduced, and the references suggested were added:

As already mentioned, Cur enhances NRF2 expression and stability (through the inhibition of KEAP1) (Shin et al., 2020) and promotes the migration of NRF2 to the nucleus, activating NRF2 downstream targets that are important to prevent oxidative stress-inhibiting inflammatory mediators in many different cellular models, such as macrophage cells (Thiruvengadam et al., 2021). Additionally, Cur treatment changed the methylation status of the NRF2 gene's CpG promoter region, causing NRF2 and its target gene, NQO-1, to be expressed again and having a chemopreventive impact on prostate cancer (Khor et al., 2011). To learn more about the molecular processes behind Cur's effects on NRF2, in silico investigations like molecular docking and molecular dynamics simulations may be important in the following steps of this research (Furlan and Bren, 2021; Tsouri et al., 2024).

Section Materials and Methods 

The references to the applied protocols should be provided in all subsections.

We added the references relative for each protocol

We would like to thank the reviewers again for the suggestions they gave us. The paper is now improved in the research design, in the description of the methods, in the presentation of the results, and in the conclusions, which are more clearly supported by the results.

We hope that the revised version of the paper will fulfill the requirements for publication in the International Journal of Molecular Science.

Please have the most cordial regards, from my collaborator and myself.

Yours sincerely,

Daniele Bottai, Ph. D.
[email protected]
University of Milan
Assistant Professor in Physiology
Department of Pharmaceutical Sciences
Section of Pharmacology and Biosciences
University of Milan
Via Balzaretti 9, 20133 Milan ITALY

Tel 0250318233

Reviewer 2 Report

Comments and Suggestions for Authors

IJMS- 3095540 comments

Spinal muscular atrophy (SMA), an autosomal recessive, progressive neurodegenerative disease, is characterized by the degeneration of α-motor neurons in the ventral grey horn of the spinal cord. The disease is associated with the inactivation (deletion or mutation) of the survival motor neuron 1 (Smn1) gene, located in the unstable telomeric region of chromosome 5.

In this manuscript, Adami et al isolated neural stem cells (NSCs) from SMN-Δ7 mice and examined the therapeutic role of curcumin in isolated NSCs, including cell proliferation, clonogenic capacity, stemness neural markers, metabolic activity, and SMN protein levels. The following concerns need to be addressed to strengthen the quality of the manuscript.

1.      The data presented in the current version are very descriptive. The authors may need to add some mechanistic data to support the acceptance of IJMS. 

2.      The data in Figures 5-6 showed that curcumin increased the nuclear/cytoplasmic ratio of NRF2, a nuclear factor that regulates the SMN gene at the transcriptional level. It is not clear if the enhanced level of nucleic NRF2 can upregulate Smn1.

3.      It is necessary to follow the guidelines of standard nomenclature for genes and proteins between human and mouse.

4.      Please add the group labeling insert in Figure S1.

5.      It is necessary to use the same indication for p values in all figures. The authors used the “*” in Figures 1-3, but “$” in Figure 4.    

Author Response

Milan 23/07/24

To whom it may concern:

First of all, we would like to express our gratitude for the helpful suggestions the editor and the reviewer gave us. We are convinced that those recommendations will allow us to substantially improve our paper.

Here you can find the changes we made to the manuscript. The changes we made are indicated in blue as insertions and in red as cuts.

We chose to transmit the edits to both reviewers since a significant adjustment was required.

Response to Reviewer 2

IJMS- 3095540 comments

Spinal muscular atrophy (SMA), an autosomal recessive, progressive neurodegenerative disease, is characterized by the degeneration of α-motor neurons in the ventral grey horn of the spinal cord. The disease is associated with the inactivation (deletion or mutation) of the survival motor neuron 1 (Smn1) gene, located in the unstable telomeric region of chromosome 5. 

In this manuscript, Adami et al isolated neural stem cells (NSCs) from SMN-Δ7 mice and examined the therapeutic role of curcumin in isolated NSCs, including cell proliferation, clonogenic capacity, stemness neural markers, metabolic activity, and SMN protein levels. The following concerns need to be addressed to strengthen the quality of the manuscript.

1.      The data presented in the current version are very descriptive. The authors may need to add some mechanistic data to support the acceptance of IJMS.

We added a new figure describing our hypothesis and also the possible effect on NRF2 upregulation of SMN (see also point 2).

2.      The data in Figures 5-6 showed that curcumin increased the nuclear/cytoplasmic ratio of NRF2, a nuclear factor that regulates the SMN gene at the transcriptional level. It is not clear if the enhanced level of nucleic NRF2 can upregulate Smn1.

We added a new figure describing our hypothesis and also the possible effect on NRF2 upregulation of SMN, and we discussed this hypothesis as indicated below (see also point 1).

Our mouse model (Le et al., 2005) contains all the SMN2 human gene, and the SMA cDNA lacks exon 7, which is driven by a 3.4 kb SMN human promoter fragment. Little differences were detected between the telomeric (SMN1) or centromeric (SMN2) promoters (Monani et al., 1999; Boda et al., 2004), so both contain the two potential AREs, but the SMN2 promoter has almost half the transcriptional activity of the telomeric promoter.

We can hypothesize with a meccanistic model that Cur could increase the levels of SMN protein, as described in Fig. 8. Cur can modify the Keap1 Cys151 residue and inhibits the ability of the Cullin3-Rbx1 E3 ubiquitin ligase complex to target Nrf2 (Shin et al., 2020). This will allow an increase in the translocation of NRF2 into the nucleus (Fig. 7 and Fig. 8 supp) and grow the binding to the ARE sequence that is present in the 3.4 kb SMN human promoter fragment or in the human SMN gene inserted in this trangenic animal (Cui et al., 2023). Consequently, Cur can induce an increase in the levels of SMN protein in SMA animals (Fig. 6). However, the small increase of SMN due to curcumin treatment could be attributed to the low efficiency of the SMN2 human promoter with respect to the SMN1 promoter (Cui et al., 2023) that is present in the SMA mouse.

On the other hand, in WT animals, the NRF2 ratio level is already quite high in comparison to SMA animals, and Cur is not significantly able to modify its translocation into the nucleus. Moreover, it is not yet clear if in the mouse Smn promoter are present AREs; the mouse Smn promoter region is quite different from the human one. By using a searching tool fo putative transcription binding site, we found that some putative ARE were present in the Smn mouse promoter, although we are not certain that these site could really bind NRF2 (https://alggen.lsi.upc.es).

.

3.      It is necessary to follow the guidelines of standard nomenclature for genes and proteins between human and mouse.

We revised all the nomemclature for genes and proteins.

https://www.biosciencewriters.com/Guidelines-for-Formatting-Gene-and-Protein-Names.aspx

4.      Please add the group labeling insert in Figure S1.

The group labeling was adde in Fig. 1 supp.

5.      It is necessary to use the same indication for p values in all figures. The authors used the “*” in Figures 1-3, but “$” in Figure 4.   

The * is relative to the comparison between WT-ctr and SMA-ctr, whereas $ is relative to the comparison between SMA-ctr and SMA-Cur. This is made in order to be consistent with the supplementary figures. 

We would like to thank the reviewers again for the suggestions they gave us. The paper is now improved in the research design, in the description of the methods, in the presentation of the results, and in the conclusions, which are more clearly supported by the results.

We hope that the revised version of the paper will fulfill the requirements for publication in the International Journal of Molecular Science.

Please have the most cordial regards, from my collaborator and myself.

Yours sincerely,

Daniele Bottai, Ph. D.
[email protected]
University of Milan
Assistant Professor in Physiology
Department of Pharmaceutical Sciences
Section of Pharmacology and Biosciences
University of Milan
Via Balzaretti 9, 20133 Milan ITALY

Tel 0250318233

Round 2

Reviewer 2 Report

Comments and Suggestions for Authors

I have no more comments.